# Hospital at Home care for older patients with cognitive impairment: a protocol for a randomised controlled feasibility trial

Maaike A Pouw,[1,2] Agneta H Calf,[1] Barbara C van Munster,[1,3] Jan C ter Maaten,[4] Nynke Smidt,[1,5] Sophia E de Rooij[1]

¹Department of Geriatrics, University Medical Center Groningen, University of Groningen, Groningen, The Netherlands
²Department of Internal Medicine, Martini Hospital, Groningen, The Netherlands
³Department of Geriatrics, Gelre Hospitals, Apeldoorn, The Netherlands
⁴Department of Internal Medicine, Emergency Department, University of Groningen, University Medical Center Groningen, Groningen, The Netherlands
⁵Department of Epidemiology, University of Groningen, University Medical Center Groningen, Groningen, The Netherlands

**Correspondence to**
Maaike A Pouw;
m.a.pouw@umcg.nl and Sophia E de Rooij;
s.e.j.a.de.rooij@umcg.nl

## ABSTRACT

**Introduction** An acute hospital admission is a stressful life event for older people, particularly for those with cognitive impairment. The hospitalisation is often complicated by hospital-associated geriatric syndromes, including delirium and functional loss, leading to functional decline and nursing home admission. Hospital at Home care aims to avoid hospitalisation-associated adverse outcomes in older patients with cognitive impairment by providing hospital care in the patient's own environment.

**Methods and analysis** This randomised, non-blinded feasibility trial aims to assess the feasibility of conducting a randomised controlled trial in terms of the recruitment, use and acceptability of Hospital at Home care for older patients with cognitive impairment. The quality of care will be evaluated and the advantages and disadvantages of the Hospital at Home care programme compared with usual hospital care. Eligible patients will be randomised either to Hospital at Home care in their own environment or usual hospital care. The intervention consists of hospital level care provided at patients' homes, including visits from healthcare professionals, diagnostics (laboratory tests, blood cultures) and treatment. The control group will receive usual hospital care. Measurements will be conducted at baseline, during admission, at discharge and at 3 and 6 months after the baseline assessment.

**Ethics and dissemination** Institutional ethics approval has been granted. The findings will be disseminated through public lectures, professional and scientific conferences, as well as peer-reviewed journal articles. The study findings will contribute to knowledge on the implementation of Hospital at Home care for older patients with cognitive disorders. The results will be used to inform and support strategies to deliver eligible care to older patients with cognitive impairment.

**Trial registration number** NTR6581; Pre-results.

## Strengths and limitations of this study

► This study addresses the feasibility of Hospital at Home care in patients with cognitive impairment, a patient population that is often excluded from participation in scientific research.
► A process evaluation facilitates the investigation of factors that influence the experiences and perceptions of all persons involved in Hospital at Home care.
► Stakeholders were involved in the development of the design of the study which will support the implementation of Hospital at Home care and a future trial.
► Because of a limited sample size due to the study being centred on feasibility, results will not show effectiveness of Hospital at Home care compared with usual hospital care.

## INTRODUCTION
### Background

An acute hospital admission is a stressful life event, particularly for older people. In addition to the stress of an acute illness, the hospital admission itself contributes to this stress.[1] Older hospitalised patients are often deprived of sleep, and they spend an average of 20 of every 24 hours in bed, they become poorly nourished, and experience sensory deprivation or overstimulation, resulting in confusion.[2–5] These adverse effects of hospitalisation contribute to the occurrence of geriatric conditions, such as delirium, functional decline, falls, incontinence, hospital acquired infections and pressure injuries.[6–9] Adverse effects of hospitalisation occur more easily in older people, particularly in those who are already frail, a growing portion of the worldwide ageing population.[10 11] Frailty is a state of increased vulnerability to external stressors resulting from ageing-associated declines in reserve and function across multiple physiological systems.[10] Cognitive impairment (ie, dementia) is an important contributor to frailty in older people.[12] Cognitively impaired older people are more likely to become hospitalised and once admitted, they experience longer stays than their peers without cognitive impairment.[13–15] The combination of hospitalisation and cognitive impairment

in older people is associated with further functional and cognitive declines and higher mortality rates, and it leads to more discharges to long-term care facilities.[16 17] The prevalence and worldwide burden of cognitive impairment will continue to increase as the average life expectancy increases.[18] The total number of people with cognitive impairment is estimated to be 75.6 million in 2030 and will nearly triple in 2050 to 135.5 million.[19] An increase in the number of hospital admissions of older people with cognitive impairment and an increase in number of hospitalisation-associated adverse outcomes are therefore to be expected.

Besides adverse outcomes of hospitalisation, many older people and their caregivers do not necessarily desire a hospital admission in case of an acute illness or exacerbation of a chronic illness. Fried *et al* (2000) have studied the preferences of community-dwelling persons 65 years of age and older who were hospitalised with a primary diagnosis of congestive heart failure, chronic obstructive pulmonary disease (COPD) or pneumonia. The authors reported that over 50% of older patients preferred to receive hospital treatment at home, because they felt that their homes were more comfortable.[20] In the treatment preferences of seriously ill patients aged 60 years and older, the likelihood of cognitive and functional impairment as an adverse outcome of the treatment was weighed in the decision-making process. There was a substantial decrease in the number of participants who opted for treatment if the likelihood of impairment after treatment was 50% or higher.[21]

Hospital at Home care could provide an effective alternative to inpatient care for a select group of elderly patients now requiring hospitalisation. Hospital at Home care is coordinated, multidisciplinary care in the homes of people who would otherwise be admitted to the hospital. Hospital at Home care is an accepted alternative to inpatient hospital level care in several countries (eg, the USA, Italy and the UK) but not yet in the Netherlands.[22] Since the 1990s, Hospital at Home has been evaluated in (older) persons with various acute medical conditions, such as heart failure, exacerbations in COPD and infections (eg, cellulitis, pneumonia).[22] In systematic reviews comparing alternative strategies to inpatient hospitalisation, lower or equal mortality rates and return hospitalisation rates (ie, subsequent admissions after discharge) were found for Hospital at Home care, there was a lower incidence of delirium, and there was a positive effect on patient and caregiver satisfaction.[22–24] Only one completed trial conducted in Italy included 109 patients with cognitive impairment (ie, dementia), Tibaldi *et al* reported a positive effect of a Hospital at Home intervention on behavioural disturbances and caregiver stress in patients with dementia.[25] Results of a still ongoing trial including people with i.a. cognitive impairment in the UK, will follow in the near future.[26]

Whether Hospital at Home care provides a suitable alternative with regard to other outcomes as patient satisfaction, quality of care, hospitalisation-associated adverse events and costs in older people with cognitive impairment remains unclear and further research is needed. Therefore, our primary aim is to investigate the feasibility of a Hospital at Home care programme for older patients with cognitive impairment in terms of the patient recruitment, use and acceptability, and second to investigate the advantages and disadvantages of Hospital at Home care compared with usual hospital care from different perspectives.

Objectives of this study are the following:
1. To assess the participation rate of the Hospital at Home trial among patients 65 years and older with cognitive impairment, acute illness and emergency hospital admission. What are the reasons for non-participation?
2. To assess the potential advantages and disadvantages of Hospital at Home care and usual hospital care for the patients, caregivers and Dutch medical health system.
3. To assess the feasibility of Hospital at Home care in terms of the quality of care with regard to geriatric syndromes, institutionalisation, mortality, total days with urinary catheter, length of stay (in the hospital or in Hospital at Home care) and timing/intensity of the contact with healthcare professionals.

## METHODS AND ANALYSIS
### Trial design
The design is a randomised controlled feasibility trial and will use a process evaluation. This study will be conducted at the medical emergency department (ED) of the academic hospital of the University Medical Center of Groningen in the Netherlands and will evaluate cognitively impaired older patients who are in need of acute hospital care. Figure 1 shows the trial design summary. Participants will be randomised to either Hospital at Home care or usual hospital care in a 4:1 ratio, respectively. Patients will be randomised using a computerised random number generator (http://www.randomization.com), including block randomisation.

An independent research nurse who is not involved in the patient care will complete the baseline assessment and allocate the participants (using sealed sequenced envelopes) into the Hospital at Home care (intervention) or usual hospital care group (control). The research nurse will not be aware of the randomisation method. The participants, healthcare professionals and research staff will not be blinded to the intervention. The reporting of the design of this trial protocol is in accordance with the SPIRIT (Standard Protocol Items: Recommendations for Interventional Trials) 2013 statement for clinical trial protocols.[27]

### Study population
Patients 65 years of age and older who are admitted to the medical ED will be identified by the ED staff as potential eligible patients. Subsequently, the ED staff will inform the research nurse. The research nurse will

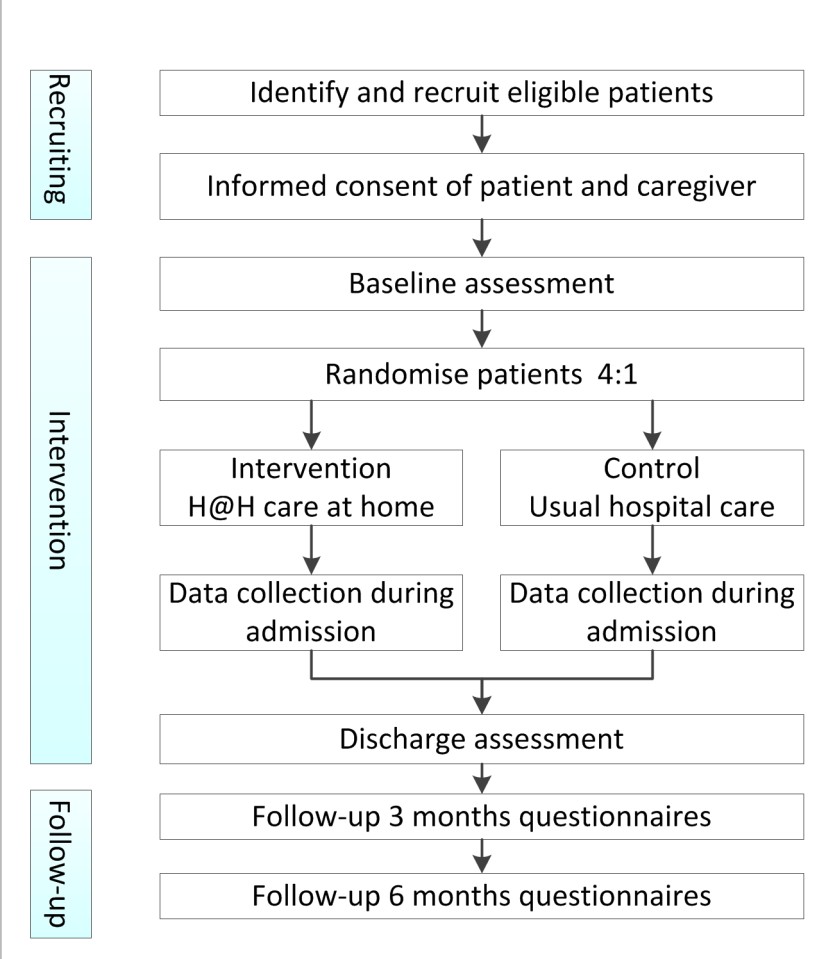

**Figure 1** Flow chart of trial design summary

complete the eligibility criteria checklist (table 1). The research nurse will ask the patient and their informal caregiver (ie, partner, child, relative, friend) for their willingness to participate in the study and to provide written informed consent. The patient and informal caregiver will need to both sign the informed consent form before the patient can participate in the H@H trial.

**Table 1** Patient eligibility criteria for participation in Hospital at Home trial

| Inclusion criteria | Exclusion criteria |
| --- | --- |
| ► Age 65 years of age and older<br>  – Cognitive impairment, that is, dementia, delirium or other cause of cognitive impairment, and either Previously diagnosed or documented in the medical records or<br>  – Identified by the ED clinician (eg, with the 4AT test and/or six-item cognitive impairment test)[42–44]<br>► Presented at the ED with a defined acute illness<br>  – Required hospital admission, according to the attending ED physician but not expected to require emergency interventions, Modified Early Warning Score ≤2 points[45–47]<br>► Living in hospital's catchment area (<25 km)<br>► Informal caregiver is present and able to understand and perform instructions and consented to participate in the trial<br>► Home suitable for Hospital at Home care (available informal caregiver, running water, adequate heating, safety)[44] | ► Previously enrolled<br>► Hospitalised within the 7 days preceding ED presentation<br>► Nursing home residents or awaiting a nursing home place on an active waiting list (excluding so-called sleeping waiting list candidates)[27]<br>  – Additional care needed: Required surgical assessment<br>  – Suspected acute coronary syndrome or cardiac arrythmia[45]<br>  – Dialysis dependent patients[45]<br>  – Expected terminal events[45] or in need of diagnostic or palliative care due to oncological or haematological illness |

ED, emergency department.

An evaluation to assess the mental capacity is conducted by the involved ED staff in the setting of the ED assessment. If the participant lacks the capacity to consent (mentally incapacitated), and an informal caregiver (ie, partner, child, friend) is present, this informal caregiver will be asked to act as a personal consultee. The personal consultee will determine whether he or she believes that participation in the study would be in accordance with the values and interests of the individual and will subsequently sign the patient's informed consent form.

### Sample size of study population

Based on the numbers available from the ED of the University Medical Center Groningen, the Netherlands, we calculated that an average of 3990 older patients 65 years and over is admitted to our medical ED each year. The Hospital at Home trial will be introduced during working hours, which provides an estimated 1900 patients per year. Not all 1900 patients will be eligible for study inclusion. Based on screening of ED medical records, approximately 15% of the patients meet the eligibility criteria for Hospital at Home care, resulting in 285 eligible persons per year. In recent randomised controlled trials (RCTs) of Hospital at Home care in Italy, 54% and 57% of the eligible patients was willing to participate and gave informed consent.[28 29] We presume a similar consent rate of 50%, as described in these previous clinical trials, and expect for 143 patients to be included.

### Study procedures

After (written) informed consent is obtained, all participants will complete two brief tests to assess cognitive impairment, and the participant and caregiver will complete the baseline assessment. Subsequently, randomisation takes place to either (1) the Hospital at Home care-intervention, translocation of care from the hospital to a participants' home or (2) the control group, usual hospital care. All care will be delivered according to hospital protocols, current regulations and guidelines and, if needed, described in the standard operating procedures (SOPs). If no informed consent is given by either the patient or the caregiver, the reasons for non-participation, date of birth, sex of the patient and the relationship between patient and caregiver will be reported.

### INTERVENTION
### Hospital at Home care

Hospital at Home care will be delivered by a multidisciplinary team consisting of a physician, nurse, pharmacist and physiotherapist. Depending on the participants' needs, other disciplines (eg, a dietician, occupational therapist or social services) can be involved in the Hospital at Home care. The day-to-day care will be provided by the nurse and physician visiting the participant. The Hospital at Home care team works under the responsibility of the medical specialist in the hospital, and 24/7 consultation

of the expertise and services of the hospital is part of the protocol.

The participants allocated to Hospital at Home care will receive hospital level care in their own homes. After a stay of one night in the hospital, while the Hospital at Home care arrangements are being made, the participant will be transferred home and receive Hospital at Home care. The Hospital at Home nurse is responsible for the day-to-day care and will be present on arrival of the participant at his/her residence. Hospital at Home care is described in the care protocols including SOPs and could include intravenous therapy (eg, antibiotics, fluid, and/or diuretics), oxygen therapy, and/or nebuliser, indwelling urine catheter or a nasopharyngeal food tube. After the care intake and a period of direct nursing supervision, the participant will receive intermittent nursing visits daily (starting with three times per day), including weekends and public holidays. The Hospital at Home physician will make a home visit every day (excluding weekends). The Hospital at Home physician and nurse will be available for emergency visits. The participant will receive a medical alert device in the house, with a 24/7 connection to an on-call service. Alert instructions will be explained to the participant and caregiver. A physiotherapist will visit the participant at home to evaluate any problems with balance and/or walking and immobility. The Hospital at Home team works under the supervision of the hospital medical specialist. Daily screenings and measurements will be recorded in a Hospital at Home record, which stays with the participant. Diagnostic procedures and therapeutics that cannot wait and are not available at home, such as endoscopy or CT scan, will be arranged through brief visits to the hospital. The participant will be 'admitted' to the Hospital at Home care for as long as indicated.

### Discharge from Hospital at Home care

If the participant recovers to such an extent that hospital level care is no longer needed, the participant will be discharged from the Hospital at Home care programme, similar to the discharge procedure when the participant would receive usual hospital care. Hospital at Home care will end with discharge planning with the participant, family, Hospital at Home physician and nurse. The discharge plan includes follow-up appointments (eg, at the hospital or general practitioner), information on medication, warning signs and symptoms and an ongoing management plan. All hospital-related care equipment will be removed from the participants' house, and arrangements with home care agencies and/or paramedical staff will be reviewed and adjusted to the current situation.

### Hospital care as usual

Participants allocated to the control arm will be admitted to a hospital ward and receive usual hospital care. After admission and intake on the ward, the participant will receive intermittent visits from the ward nurse multiple

times a day. The ward physician will visit once every day (excluding weekends), with extra visits provided if needed. An emergency alert device, through which nurses and physician can be contacted, will be placed next to the bed. A physiotherapist will visit the participant at the ward to address problems with balance and/or walking and immobility. Depending on the participants' needs, other disciplines (eg, a dietician, occupational therapist or social services) can be involved in the hospital care. The medical record is the hospital record and additional H@H research forms will be added to this record for research purposes. The participant will be admitted to the hospital for as long as indicated.

### Hospital discharge

If the participant recovers to such an extent that hospital level care is no longer needed, the participant will be discharged from the hospital after discharge planning with the participant, family, physician and nurse. The discharge plan includes follow-up appointments (eg, at the hospital or general practitioner), information on medication, warning signs and symptoms and an ongoing management plan. Arrangements with home care agencies and/or paramedical staff will be reviewed and adjusted to the current situation.

### Follow-up

At 3 and 6 months following randomisation, all participants will be contacted for an interview by telephone[30] or a face-to-face interview, if needed. The participants will be allowed to receive support with these questions from their relatives or informal caregivers. An interview will require a maximum of 30 min. In case of institutionalisation or mortality, this event will be recorded. Additionally, information on hospital readmission and length of stay will be collected from the hospital administration system and health insurers. Mortality and nursing home placements will be collected from registries from the general practitioner and municipalities.

### Timing of measurements and outcome measures

Data will be collected at baseline at the ED, during admission (in Hospital at Home or hospital), at discharge and at 3 and 6 months following randomisation, plus or minus 2 weeks. An overview of the timing of measurements and outcome measures are shown in table 2.

### Feasibility

For the participation rates, the proportion of participants per step will be calculated. The reasons for non-participation and data concerning the characteristics of non-participants will be collected. We consider the participation rate feasible when it is similar to the participation rate as is described in previous RCTs and around 50% of the eligible patients will consent to participate.[28 29] Quality of care will be measured by collecting data on patient, institutionalisation (eg, to the hospital or nursing home), mortality, activities of daily living functioning, prevalence of hospitalisation-associated geriatric syndromes, the

length of stay in the hospital or Hospital at Home care programme and contact with healthcare professionals. The study is considered feasible if the quality of care of Hospital at Home care on these measurements is non-inferior to usual hospital care.

### Other outcomes measures

Advantages and disadvantages of the Hospital at Home care programme will be assessed through multiple instruments and questionnaires. The instruments are validated and used in community-dwelling older patients with cognitive disorders. Additional data will be collected on the time spent at home (home time); total number of days alive and out of the hospital or a skilled nursing facility in the 6 months following the randomisation at the ED,[31] the number of transfers (home↔hospital) and the number of healthcare professionals involved.

Cost data will be collected, as described by Drummond et al, including the costs to the healthcare system, patients and families, and other sectors.[32] The volume of care use will be extracted from hospital files and combined with the reference cost values, as provided by the cost guidance module of the Dutch National Health Care Institute.[33]

### Process evaluation

A process evaluation will be conducted as part of the feasibility study to understand the barriers and facilitators to participate and to gain an understanding of the experiences and perceptions of Hospital at Home care of participants and healthcare professionals. From all eligible patients who declined to participate, data concerning the patient characteristics and reasons for non-participation will be collected. At the end of the trial, a representative sample of participants and/or their informal caregivers will be invited for an interview to evaluate their experiences receiving Hospital at Home care. The interviewer will not be a an active member of the research team or involved in day-to-day care and will explore independently how the participants perceived Hospital at Home care, including the contact with the health professionals and the impact of Hospital at Home care on their lives and their caregivers' personal lives. In case of participant dropout, efforts will be made to obtain an understanding of why the participants did not complete the trial.

In addition, a representative sample of healthcare professionals, consisting of physicians and nurses working in the ED, physicians and nurses providing the Hospital at Home care, and general practitioners, will be asked to participate in a face-to-face interview. The healthcare professionals will be asked about their experiences and opinions about the H@H trial and Hospital at Home care. All interviews will be transcribed verbatim, and a framework analysis will be used as the method of qualitative data analysis.[34]

### Data management

All data will be entered in an electronic trial-specific database, with the participants identified by a unique trial number. Confidentiality of participant information

**Table 2** Overview of the content and description of outcome measures and timing of measurements

| | | Timing of measurements | | | | | |
| --- | --- | --- | --- | --- | --- | --- | --- |
| | Description and instrument | Screening | Baseline | Admission | Discharge | 3 months | 6 months |
| Degree of illness based on physiological parameters | Vital signs alarm score; Modified Early Warning Score[48] | R | | | | | |
| Cognitive impairment | 4AT test for delirium*, six-item cognitive impairment test for cognitive impairment*[42 49] | R | | | | | |
| Sociodemographics | Date of birth, nationality, household composition, marital status, highest level of education | | R | | | | |
| Health status | Charlson Comorbidity Index*[50] | | R | | | | |
| Identifying at-risk patients | Safety management system patient screening (VMS)[51] | | R | | | | |
| Functional status | Activities of daily living (ADL), modified Katz-ADL index score[52] | | R | | R | R | R |
| Health status | EuroQol-5D-5L*[53] | | R | | R | R | R |
| (Health-related) quality of life, well-being | Icepop capability measure for Older people (ICECAP-O)*[54] | | R | | R | R | R |
| Caregiver burden | Self-rated burden scale*, caregiver strain index*[55 56] | | C | | C | C | C |
| Medical consumption | Imta Medical Consumption Questionnaire (imcq)*[57] | | P,C | | | P,C | P,C |
| Hospitalisation-associated geriatric syndromes | Infections, falls, pressure injuries, in case of delirium; delirium observation scale score (DOSS)[58] and use of physical or chemical restraints, total days with a urinary catheter | | | N | | | |
| Nutrition | Malnutrition Universal Screening Tool (MUST)[59] | | N | | | | |
| | Food intake, fluid intake | | | N | | | |
| Pain | Numeric Rating Scale-score (NRS) for pain[60] | | | N | | | |
| Health perception | (Rotterdam) symptom checklist*[61] | | N | | | | |
| Immobility | Hierarchical assessment of balance and mobility (HABAM)[62] | | | Ph | | | |
| Satisfaction with care | Client Satisfaction Questionnaire 8 (CSQ-8)*, care evaluation question*[63 64] | | | | P<br>P, C, N, D | | |
| Mortality | Mortality at 30 days, 3 months and 6 months after baseline* | | | | N | R | R |
| (Re)admission hospital | Length of stay, readmission rate at 30 days, 3 months and 6 months after baseline* | | | | N | R | R |

Assessed by: C, caregiver; D, doctor; N, nurse; P, participant; Ph, physiotherapist; R, research nurse.
*All assessments are extra for trial purposes and are not part of the medical treatment.

will be maintained throughout the trial. Information can only be traced to the participants by designated researchers. The database will be stored and maintained by Castor Electronic Data Capture, compliant with GCP guidelines and the European Data Protection Directive (Castor Electronic Data Capture, 2017; Ciwit BV, Amsterdam, the Netherlands). Data will be stored for a maximum period of 15 years after the study has ended, according to Dutch law.[35]

### Statistical analysis

The participant flow diagram, according to Consolidated Standards of Reporting Trials (CONSORT) guidelines,[36] will provide a summary of the recruitment and declination rates in percentage (%) at baseline, discharge and 3-month and 6-month follow-ups. Distributions of the data at baseline, discharge and 3 and 6 months after randomisation will be explored, with unusual values noted and explained. Variables will be summarised as the n (%), mean (SD) or median (IQR) for each group to characterise the sample and search for any imbalances. The percentages, means and SD, and medians and IQRs will be calculated to describe the quality of care and the advantages and disadvantages of Hospital at Home care at baseline, discharge and 3-month and 6-month follow-ups.

## Monitoring and participant safety

Although, the H@H trial is considered to be a low risk trial, the participant safety will be monitored by an independent Data Monitoring Committee (DMC). The DMC will consist of two members: an experienced clinician and an epidemiologist. Members of the DMC are independent of the trial and will discuss each individual participant with serious adverse events. The DMC will receive and review the serious adverse events and evaluate the risk involved with negative outcomes. The DMC is authorised to make recommendations to temporary put on hold or ending the study prematurely when participant safety is an issue, based on their findings. All serious adverse events will be reported to the principal investigator within 24 hours of knowledge of the event and then subsequently reported to the Dutch portal for medical research involving human subjects.

## DISCUSSION

Reducing unwanted hospital admissions in older patients with cognitive impairments and facilitating patient-centred care in a patient's preferred location is a goal worthy of pursuing. This goal aligns with the tenet of the current Dutch government and the advice provided by the Dutch Council for the Environment and Infrastructure: to actively promote and enable people to live independently in their own homes for as long as they desire.[37] Previous trials and a recent review have confirmed that alternative management strategies for low-risk patients with acute medical conditions conventionally treated through hospitalisation exist with positive impact on patient satisfaction, are effective and can be safely achieved in lower cost settings.[22 23 38]

Introducing a Hospital at Home care trajectory in the Netherlands is incited by the principles of value-based healthcare: improving the patients experience of care and as a result of this process reducing the costs.[39] All countries with an ageing population experience pressure, in terms of shortage of (emergency) hospital beds and rising healthcare costs. Hospital at Home care could be shown beneficial in facilitating higher valued care for patients and their caregivers without additional costs. Benefit should be measured in other outcomes than clinical indicators such as mortality. To illustrate, one of the outcomes of a future RCT could be the time spent at home. Time spent at home has been defined as the total number of days alive and out of the hospital or a skilled nursing facility in the 6 months after hospital admission.[31] It has been used as a primary outcome in a follow-up study of older patients with acute hospital admissions, and has been demonstrated to be of more importance in older patients.[17 40 41] Evaluation of time spent a home in this feasibility study could support estimating a sample size based on a patient-relevant outcome in a future RCT.

This study will be the first to investigate the feasibility of providing acute hospital care at home for older patients with cognitive impairment in the Netherlands. Studying Hospital at Home care and identifying the barriers and facilitators will support the implementation of Hospital at Home care and break new ground for a future RCT investigating the (cost-)effectiveness.

## ETHICS AND DISSEMINATION

The trial will be conducted in accordance with the Declaration of Helsinki 1996, principles of good clinical practice and the University Medical Center of Groningen Research Code. Any protocol amendments will be submitted to the ethics committee. A register of the protocol amendments will be available in the study protocol.

The results of the trial will be reported according to the CONSORT guidelines and will contribute to knowledge of the implementation of Hospital at Home care and patient-centred acute care for older patients with cognitive impairment. The study will also contribute to the knowledge of the transmural cooperation and costs of providing care, in terms of the translocation of hospital care to home. Regularly updates will be published on the study website and in newsletters. Conferences and meetings will be held for all involved healthcare professionals. Participants who requested information on the study will be sent a lay summary. A publication policy will be agreed on with co-applicants. The study findings will be published in relevant peer-reviewed journals.

**Acknowledgements** The authors thank all care professionals of Alzheimer Nederland, Espria, Icare, TSN, Buurtzorg, umcGroningen thuis, MCZ, Netwerk Dementie Groningen, Netwerk Dementie Drenthe, Gemeente Groningen, WIJ Groningen, Zorgbelang, Healthy Aging Network Northern Netherlands, Stichting Effectieve Ouderenzorg, ZonMW, Deltaplan Dementie, Nationaal Programma Ouderenzorg, Zorg innovatie Forum and the elderly of Denktank 60+ Noord for their time, critical appraisal and advise.

**Contributors** MAP contributed to the conception and design of the protocol, drafted the manuscript and revised the final manuscript. AHC, BCvM, JCtM and NS made substantial contributions to the conception and design of the protocol. NS contributed to the methodological aspects of the protocol. AHC, BCvM and JCtM contributed to the clinical aspects of the protocol. SEdR conceived the study, developed the protocol together with MAP and wrote the funding applications. All authors critically revised the manuscript and approved the final version of this manuscript.

**Funding** This study is an independent research funded by ZonMw (project Memorabel, Deltaplan Dementie), project no: 733050401 (contact: memorabel@zonmw.nl). Additional funding is provided by the Dutch Ministry of Health, Welfare and Sport (Ministerie van Volksgezondheid, Welzijn en Sport) and the University Medical Center Groningen.

**Competing interests** None declared.

**Patient consent** Detail has been removed from this case description/these case descriptions to ensure anonymity. The editors and reviewers have seen the detailed information available and are satisfied that the information backs up the case the authors are making.

**Ethics approval** This study obtained ethical approval from the Medical Research Ethics Committee of the University Medical Center Groningen, April 2017 (ref: 2016.686).

**Provenance and peer review** Not commissioned; externally peer reviewed.

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
