## [Reviewer comments · BMJ Open]

ARTICLE DETAILS

TITLE (PROVISIONAL)	Hospital at Home care for older patients with cognitive impairment: A protocol for a randomized controlled feasibility trial
AUTHORS	Pouw, Maaïke; Calf, Agneta; van Munster, Barbara; ter Maaten, Jan; Smidt, Nynke; De Roolj, Sophia

VERSION 1 – REVIEW

REVIEWER	rebekah Schiff Dept Ageing and Health Guys and St Thomas NHS Foundation Trust UK
REVIEW RETURNED	23-Nov-2017

GENERAL COMMENTS	Looking at the feasibility of hospital at home for people with cognitive impairment is an important question. However the objectives of this study appear to be wider than this; Objectives 1 and 3 are in line with the title but 2 looks at comparing hospital vs hospital at home in terms of quality of care e.g catheter use. I believe this is a different question and would need a much larger sample size. However data from this study could inform this sample size calculation. A missing reference is Trials. 2017 Oct 23;18(1):491. doi: 10.1186/s13063-017-2214-y. A multi-centre randomised trial to compare the effectiveness of geriatrician-led admission avoidance hospital at home versus inpatient admission. This study is including people with cognitive impairment but in a UK setting. Methodology; 1. I think defining cognitive impairment might need to be a bit tighter as medical records can often be inaccurate- so perhaps a cognitive assessment on all at screening for the study.2. A News score of less than 2 suggests these are very physically well people- if this intervention is truly replacing hospital care perhaps higher scores should be considered as well?3. Your multidisciplinary Hospital at home seems to have no pharmacist input which is often necessary in such schemes4. Management of behavioural issues in delirium/dementia is often a reason for people to end up being admitted for otherwise minor new physical issues- how will this study seek to involve these types of patients and support them and their families- this would be a useful question to answer and add a new dimension to hospital at home5. I was unclear why once randomised people would spend a day still in hospital before home- why not home same day? also risks a further drop out rate
--

	6. Outcome measures; some of these appear to be being done daily when i don't think changes over this time frame are clinically likely e.g the MUST score This feasibility trial will answer important questions around including cognitively impaired individuals in such trials and should be supported. Good luck.
--	---

REVIEWER	Caplan, Gideon Prince of Wales Hospital, Australia
REVIEW RETURNED	29-Nov-2017

GENERAL COMMENTS	This is an interesting and well written manuscript Major Issues  1. I understand that changes cannot be made to the protocol, but just note that I think it's a shame not to include nursing home residents in this trial, as they have the most to benefit from Hospital at Home care 2. I note that the second objective is to assess differences in quality of care, with particular regard to geriatric syndromes, and I note that two of the authors are experts in delirium, and that reduced incidence of delirium is one of the most interesting features of Hospital at Home care, (see a study that looked at geriatric syndromes in HaH: Caplan GA, Ward JA, Brennan N, et al. Hospital in the Home: a randomised controlled trial. Medical Journal of Australia 1999; 170: 156-160; and a review of delirium in HaH: Caplan GA. Does Hospital in the Home treatment prevent delirium? Aging Health 2008; 4(1): 69-74.) Given that about half of delirium cases are missed without routine screening, I am concerned that in this trial, it seems that delirium will only be screened for on admission. 3. This study is focused on people with cognitive impairment, so I would assume that a high proportion will lack the capacity to consent. The authors have not included any mechanism they will use to assess capacity. Minor Issues  1. P.8 line 35 it says the nurses will visit daily, it might be worth specifying that this includes weekends and public holidays, if that is indeed the case, which is vital in running a Hospital at Home.
---

REVIEWER	Miquel Àngel Mas Universitat Autònoma Barcelona, Catalonia
REVIEW RETURNED	01-Dec-2017

GENERAL COMMENTS	Thanks for sending this interesting work. It is an excellent study protocol (RCT feasibility trial) on admission avoidance hospital-at-home in patients with elevated risk of negative outcomes during conventional hospitalisation due to the condition of cognitive impairment. This paper will open a window in the evaluation of alternatives to hospitalisation in geriatrics. The paper is very well written. It is very clear. All parts are very well explained. I have some questions from Methods:  1. What is the reason why you randomise 4:1?
--

	2. Why do you choose a sample of 143 patients? Is the unique reason for it the period of recruitment of one year? 3. If there is a sample of patients with cognitive impairment, why do you exclude patients from Nursing Homes? Care homes are full of patients with cognitive disorders, and to avoid admission in the hospital should be a research priority in integrated care schemes as hospital-at-home...Could you, please, give an explanation on that? 4. Do you include patients with terminal dementia? Please, clarify it, with measurements to categorise it, if possible. 5. In table 2. I wonder why mortality and (re)admission to hospital are assessed by nurses at discharge. Are these nurses involved in the study intervention? I would like you to review reference list (ex. ref 22 incomplete). When you review the recent HaH literature, some schemes included patients with cognitive impairment in their strategies. You could consider including them, if you think it is relevant. I will like to thank you for the development of this study. It will help us, experts in hospital in the home interventions, to spread the evidence on hospital-based care in the community in the most vulnerable patients. Finally, I was fascinated for the use of the concept "time spent at home" in your work, I think this is a key concept in the evaluation of this scheme. We will follow your results on that. Very good job. Congratulations.
--	--

VERSION 1 – AUTHOR RESPONSE

REVIEWER 1: Rebekah Schiff

Institution and Country: Dept Ageing and Health Guys and St Thomas NHS Foundation Trust UK

General comments

1.1 Reviewer: Looking at the feasibility of hospital at home for people with cognitive impairment is an important question. However the objectives of this study appear to be wider than this; Objectives 1 and 3 are in line with the title but 2 looks at comparing hospital vs hospital at home in terms of quality of care e.g catheter use. I believe this is a different question and would need a much larger sample size. However data from this study could inform this sample size calculation.

1.1 Response: We agree with the reviewer that objective 2 is less consistent with the title and the aim of assessing feasibility. We have changed the order of objectives 2 and 3 and we have rephrased objective 3 (former 2). See revised manuscript, Page 4, Lines 110-118.

1.2 Reviewer: A missing reference is *Trials*. 2017 Oct 23;18(1):491. doi: 10.1186/s13063-017-2214-y. A multi-centre randomised trial to compare the effectiveness of geriatrician-led admission avoidance hospital at home versus inpatient admission. This study is including people with cognitive impairment but in a UK setting.

1.2 Response: We thank the reviewer for notifying us of this publication. We overlooked this as this article was published on the 23rd of October 2017 and this manuscript submitted on the 26th of

October 2017. This study will certainly add valuable information to the knowledge of hospital at home care as this study is including people with cognitive impairment as well. We have added a referral in the text and added the reference to the list of references. See revised manuscript, Page 3, Lines 99-100.

2. Methodology

2.1 Reviewer: I think defining cognitive impairment might need to be a bit tighter as medical records can often be inaccurate- so perhaps a cognitive assessment on all at screening for the study.

2.1 Response: We agree with the reviewer medical records often lack detailed information on cognitive impairment. Therefore all included patients will receive a global cognitive assessment, based on the 4AT test for delirium and the Six Item Cognitive impairment test. See revised manuscript, Page 6, Lines 168-169. There is a growing interest in administering cognitive and geriatric assessments in the Dutch emergency care and we hope in the near future all patients 65 years of age and older will receive a (short) cognitive assessment as part of their Emergency Department routine care assessment.

2.2 Reviewer: A MEWS score of less than 2 suggests these are very physically well people- if this intervention is truly replacing hospital care perhaps higher scores should be considered as well?

2.2 Response: We agree with the reviewer that people with a MEWS-score of 2 or less are physically well. We've evaluated literature on vital signs warning scores in hospitalized patients and evaluated patient records of consecutive patients attending our Emergency Department and decided to opt for a MEWS score of 2 or less to avoid discussions on patient safety. Providing acute hospital care in the home setting is a new care concept in the Netherlands and safety is one of the often mentioned concerns in replacing hospital care. We hope we can extend Hospital at Home care to people with a higher MEWS-score, after excluding safety issues in this study, in the future.

2.3 Reviewer: Your multidisciplinary Hospital at home seems to have no pharmacist input which is often necessary in such schemes

2.3 Response: We agree with the reviewer that input of a pharmacist is of importance. A pharmacist is involved in our multidisciplinary Hospital at Home care, we have adjusted this in the manuscript. See revised manuscript, Page 7, Line 179.

2.4 Reviewer: Management of behavioural issues in delirium/dementia is often a reason for people to end up being admitted for otherwise minor new physical issues- how will this study seek to involve these types of patients and support them and their families- this would be a useful question to answer and add a new dimension to hospital at home

2.4 Response: We acknowledge that management of behavioral issues in delirium and dementia is of concern, but strongly feel these people would benefit most from being treated at home instead of in an unfamiliar surrounding as a hospital setting. We expect the treatment of the underlying cause will improve the patients mental condition and aim to provide support at home. Support will consist of providing education to family and caregivers, support of dementia case managing, organization of care arrangements and providing a temporarily 24 hour care system if needed. At the end of the trial informal caregivers will be invited for an interview to evaluate their experiences receiving Hospital at Home care. We will address the concerns on behavioral issues in these interviews.

2.5 Reviewer: I was unclear why once randomised people would spend a day still in hospital before home- why not home same day? also risks a further drop out rate

2.5 Response: We agree with the reviewer it would be better for the participants and the risk on drop outs if once people are randomised they would be able to go home the same day. Unfortunately this could not be realized with regard to logistical reasons; i.e. the needed health care arrangements at home, e.g. medication preparations, delivery of medical devices in the home. We hope the results of

this feasibility trial will allow people to go home with Hospital at Home care the same day in the near future.

2.6 Reviewer: Outcome measures; some of these appear to be being done daily when i don't think changes over this time frame are clinically likely e.g the MUST score

2.6 Response: We agree with the reviewer the MUST score does not change over the intended time frame. The MUST score will be assessed only once at the start of Hospital at Home care, this was not clear in the provided Table 2 and we have now adjusted this in the table in the manuscript. In the timing of measurements in Table 2 we have changed 'daily' to 'admission'. See revised manuscript, Page 10, Line 261.

2.7 Reviewer: This feasibility trial will answer important questions around including cognitively impaired individuals in such trials and should be supported. Good luck.

2.7 Response: We thank the reviewer for this comment.

REVIEWER 2: G Caplan

Institution And Country: Prince Of Wales Hospital, Australia

1. General comments

1.1 Reviewer: This is an interesting and well written manuscript

1.1 Response: We thank the reviewer for this comment.

2. Major Issues

2.1 Reviewer: I understand that changes cannot be made to the protocol, but just note that I think it's a shame not to include nursing home residents in this trial, as they have the most to benefit from Hospital at Home care

2.1 Response: Thank you for this comment, we very much agree with the reviewer nursing home residents could benefit the most. However, with regard to interpreting the results of the current study, bias could occur due to community dwelling elderly being a heterogeneous population and including nursing home residents would add to this heterogeneity. In a larger study a subgroup analysis could be done, but unfortunately our current project and the related funding is focused on community-dwelling elderly. We sincerely hope we can extend Hospital at Home care to include nursing home residents in the near future as well.

2.2 Reviewer: I note that the second objective is to assess differences in quality of care, with particular regard to geriatric syndromes, and I note that two of the authors are experts in delirium, and that reduced incidence of delirium is one of the most interesting features of Hospital at Home care, (see a study that looked at geriatric syndromes in HaH: Caplan GA, Ward JA, Brennan N, et al. Hospital in the Home: a randomised controlled trial. Medical Journal of Australia 1999; 170: 156-160; and a review of delirium in HaH: Caplan GA. Does Hospital in the Home treatment prevent delirium? Aging Health 2008; 4(): 69-74.) Given that about half of delirium cases are missed without routine screening, I am concerned that in this trial, it seems that delirium will only be screened for on admission.

2.2 Response: Thank you for the comment and directing us towards the very relevant literature on this topic. We have added a referral to the association of Hospital at Home care with lower incidence of delirium. See revised manuscript, Page 3, Lines 95-96. In addition to the screening for delirium at admission there will be a daily judgment of the patient by the nurse involved in the day-to-day Hospital at Home care by use of the Delirium Observation Screening Scale (DOSS) (see Table 2). See revised manuscript, Page 10, Line 261.

2.3 Reviewer: This study is focused on people with cognitive impairment, so I would assume that a high proportion will lack the capacity to consent. The authors have not included any mechanism they will use to assess capacity.

2.3 Response: The reviewer is correct to point out that assessing capacity to consent is very important in people with cognitive impairment, and the mechanism used to assess is not specifically mentioned in our manuscript. When people attend the Emergency Department of our hospital, the evaluation of the mental capacity is done by the involved ED-clinician. This appraisal is obligated with regard to informed consent in treatment decisions. Approaching eligible persons to participate in the study takes place after the emergency department assessment and the assessment of capacity will already have been performed by then. We have added this information to the manuscript. See revised manuscript, Page 5, Lines 146-147.

3. Minor Issues

3.1 Reviewer: P.8 line 35 it says the nurses will visit daily, it might be worth specifying that this includes weekends and public holidays, if that is indeed the case, which is vital in running a Hospital at Home.

3.1 Response: In accordance with the reviewers' suggestion, we have specified the daily visits by the nurse include weekends and public holidays. See revised manuscript, Page 7, Line 193.

REVIEWER 3: Miquel Àngel Mas

Institution And Country: Universitat Autònoma Barcelona, Catalonia, Spain

1. General comments

1.1 Reviewer: Thanks for sending this interesting work. It is an excellent study protocol (RCT feasibility trial) on admission avoidance hospital-at-home in patients with elevated risk of negative outcomes during conventional hospitalization due to the condition of cognitive impairment. This paper will open a window in the evaluation of alternatives to hospitalization in geriatrics. The paper is very well written. It is very clear. All parts are very well explained.

1.1 Response: We thank the reviewer for this comment.

2. Methods

2.1 Reviewer: What is the reason why you randomise 4:1?

2.1 Response: Providing acute hospital care in the home setting is a new concept in the Netherlands. Whether Hospital at Home care will be feasible will depend to a great extent on the acceptance of Hospital at Home care in Dutch patients. Considering this is a new care concept, it is possible people will not opt for Hospital at Home care because they prefer care as usual (hospital care). In the ideal situation all patients who prefer Hospital at Home care would receive Hospital at Home care. However, to be able to generalize results to the general population a randomization procedure is needed. Patients preferring Hospital at Home care might substantially differ from patients opting for usual hospital care. Therefore, part of the patients who prefer to receive hospital at home care will receive usual hospital care to make the comparison possible. The randomization of 4:1 is chosen to provide Hospital at Home care for as many patients as possible and still be able to apply the results to the general population.

2.2 Reviewer: Why do you choose a sample of 143 patients? Is the unique reason for it the period of recruitment of one year?

2.2 Response: The sample of 143 patients is a convenience sample based on the number of eligible patients attending the Emergency Department of our hospital within a year.

2.3 Reviewer: If there is a sample of patients with cognitive impairment, why do you exclude patients from Nursing Homes?

Care homes are full of patients with cognitive disorders, and to avoid admission in the hospital should be a research priority integrated care schemes as hospital-at-home...Could you, please, give an explanation on that?

2.3 Response: Thank you for this comment, we very much agree with the reviewer nursing home residents could benefit the most. However, with regard to interpreting the results of the current study, bias could occur due to community dwelling elderly being a heterogeneous population and including nursing home residents would add to this heterogeneity. In a larger study a subgroup analysis could be done, but unfortunately our current project and the related funding is focused on community-dwelling elderly. We sincerely hope we can extend Hospital at Home care to include nursing home residents in the near future as well.

2.4 Reviewer: Do you include patients with terminal dementia? Please, clarify it, with measurements to categorise it, if possible.

2.4 Response: We will include community-dwelling patients with cognitive impairment, i.e. dementia, delirium or other cause of cognitive impairment, previously diagnosed or identified by the ED-clinician. The severity of the dementia will not be assessed with measurements to categorize. Most of the patients with terminal dementia in the Netherlands live in nursing homes with a mean length of stay from 4 to 6 months. We therefore assume eligible patients will probably have mild to moderate dementia.

2.5 Reviewer: In table 2. I wonder why mortality and (re)admission to hospital are assessed by nurses at discharge. Are these nurses involved in the study intervention?

2.5 Response: The reviewer is correct to point out that the assessment of mortality and (re)admission should be assessed by a research nurse, similar to the assessment at the 3 months and the 6 months follow-up. The attending nurse (N) as mentioned in the table 2 is involved in providing the Hospital at home care. We have now adjusted this in the table 2 in the manuscript. See revised manuscript, Page 10, Line 261.

2.6 Reviewer: I would like you to review reference list (ex. ref 22 incomplete). When you review the recent HaH literature, some schemes included patients with cognitive impairment in their strategies. You could consider including them, if you think it is relevant.

2.6 Response: We thank the reviewer for directing us to the reference list of the review by Shepperd et al. 2016. We have reviewed the reference list and found several studies where patients with cognitive impairment could have been included, for instance because they have excluded patients with severe dementia (Ricauda, 2008; Mendoza 2009), an altered mental status (Talcott, 2011) or with a Mini Mental State score <7 points (Davies, 2000). Although the results of these studies are most useful for our trial, we have chosen not to add them individually, other than mentioned in the review because these studies did not specifically address patients with cognitive impairment, except for the study of Tibaldi (2004). We have added a reference to the recently published trial protocol of an ongoing trial mentioned in the review of Shepperd et al.: *Trials*. 2017 Oct 23;18(1):491. doi: 10.1186/s13063-017-2214-y. A multi-centre randomised trial to compare the effectiveness of geriatrician-led admission avoidance hospital at home versus inpatient admission. This study protocol mentions specifically including patients presenting with delirium or with a background of dementia. See revised manuscript, Page 3, Lines 99-100.

2.7 Reviewer: I will like to thank you for the development of this study. It will help us, experts in hospital in the home interventions, to spread the evidence on hospital-based care in the community in the most vulnerable patients. Finally, I was fascinated for the use of the concept "time spent at home" in your work, I think this is a key concept in the evaluation of this scheme. We will follow your results on that. Very good job. Congratulations.

2.7 Response: We thank the reviewer for this comment.

VERSION 2 – REVIEW

REVIEWER	Gideon Caplan Prince of Wales Hospital Australia
REVIEW RETURNED	28-Jan-2018

GENERAL COMMENTS	The modifications to the manuscript render it satisfactory for publication, in my view.
---

REVIEWER	Miquel Àngel Mas Universitat Autònoma Barcelona, Catalonia
REVIEW RETURNED	09-Jan-2018

GENERAL COMMENTS	Thanks for your answers to the questions and for introducing the modifications in the new version of the draft. Regards
---